# Early High-Dose Methylprednisolone Therapy Is Associated with Better Outcomes in Children with Acute Necrotizing Encephalopathy

**DOI:** 10.3390/children9020136

**Published:** 2022-01-20

**Authors:** Han-Pi Chang, Shao-Hsuan Hsia, Jainn-Jim Lin, Oi-Wa Chan, Chun-Che Chiu, En-Pei Lee

**Affiliations:** 1Division of Pediatric Critical Care Medicine, Department of Pediatrics, Chang Gung Memorial Hospital, Taoyuan 33305, Taiwan; blueahbi7903@cgmh.org.tw (H.-P.C.); tw1picu@gmail.com (S.-H.H.); lin0227@adm.cgmh.org.tw (J.-J.L.); oiwamail@gmail.com (O.-W.C.); chiuchunche@gmail.com (C.-C.C.); 2College of Medicine, Chang Gung University, Taoyuan 33305, Taiwan

**Keywords:** methylprednisolone, neurologic outcomes, mortality, children, acute necrotizing encephalopathy

## Abstract

Background: The neurologic outcomes of acute necrotizing encephalopathy (ANE) are very poor, with a mortality rate of up to 40% and fewer than 10% of patients surviving without neurologic deficits. Steroid and immunoglobulin treatments have been the most commonly used options for ANE, but their therapeutic efficacy is still controversial. Method: We retrospectively reviewed the medical records of 26 children diagnosed with ANE. We also divided these patients into two groups: 21 patients with brainstem involvement and 8 patients without brainstem involvement. Pulse steroid therapy (methylprednisolone at 30 mg/kg/day for 3 days) and intravenous immunoglobulin (2 g/kg for 2–5 days) were administered to treat ANE. Results: The overall mortality rate was 42.3%, and patients who did not survive had significantly higher initial lactate and serum ferritin levels, as well as higher rates of inotropic agent use with brainstem involvement. There were no significant differences in the outcomes of pulse steroid therapy or pulse steroid plus immunoglobulin between survivors and non-survivors. When analyzing the time between symptom onset and usage of pulse steroid therapy, pulse steroid therapy used within 24 h after the onset of ANE resulted in significantly better outcomes (*p* = 0.039). In patients with brainstem involvement, the outcome was not correlated with pulse steroid therapy, early pulse steroid therapy, or pulse steroid therapy combined with immunoglobulin. All patients without brainstem involvement received “early pulse methylprednisolone” therapy, and 87.5% (7/8) of these patients had a good neurologic outcome. Conclusion: Pulse steroid therapy (methylprednisolone at 30 mg/kg/day for 3 days) administered within 24 h after the onset of ANE may be correlated with a good prognosis. Further studies are needed to establish a consensus guideline for this fulminant disease.

## 1. Introduction

Acute necrotizing encephalopathy (ANE) was first described by Mizuguchi in 1979 [1], and the hallmarks of its neuroradiologic presentations were noted as multiple and symmetrical brain lesions, including in the white matter, brain stem, thalamus, tegmentum, and cerebellum, detected on computed tomography or magnetic resonance imaging [2]. The clinical course of ANE often begins with a prodromal stage that includes fever, flu-like symptoms, and gastroenteritis, and then progresses to acute encephalopathy, with symptoms such as seizures, focal neurological deficits, and coma. The neurologic outcomes of ANE are very poor, with a mortality rate of up to 40% and fewer than 10% of patients surviving without neurological deficits [3]. 

The etiology and pathogenesis of ANE remain unclear. ANE has been reported to be associated with viruses, with the most common being influenza A and B and HHV-6 [4]. The prevailing hypothesis for the pathogenesis of ANE is that hyper-cytokinemia is triggered by viral infections [3,5]. Yoshinori et al. reported the elevation of interleukin- (IL)-6 in influenza virus–associated encephalopathy and hypothesized that the plasma concentration of IL-6 could be a predictor of prognosis [5]. Other varieties of cytokine have also been detected in peripheral blood, such as tumor necrosis receptor factor–α, IL-10, IL-15, and IL-1𝛽 [6]. Moreover, hyper-cytokinemia may result in systemic inflammation and can cause multiple organ failure.

To date, there has been no consensus on the best treatment options for ANE. Immunomodulatory therapy that suppresses cytokine production is commonly used based on the pathogenesis of hyper-cytokinemia. Steroid and immunoglobulin treatments have been the most commonly used treatment options for ANE, but their therapeutic efficacy is still controversial [3,7].

Therefore, we aimed to analyze the efficacy of steroid and immunoglobulin treatment in ANE and hypothesized that steroid and immunoglobulin treatment both contribute to better outcomes.

## 2. Materials and Methods

### 2.1. Patients

We retrospectively reviewed the medical records of 29 patients diagnosed with ANE and admitted to the pediatric intensive care unit in a tertiary hospital between January 2001 and January 2020. The diagnosis of ANE was based on the criteria established by Mizuguchi [2]. All included patients experienced acute encephalopathy, and the neuroradiological findings included symmetric and multifocal brain lesions that involved the bilateral thalami, periventricular white matter, internal capsule, putamen, cerebellum, or upper brain stem tegmentum [8]. All patients were evaluated by a pediatric neurologist. This study was approved by the Institutional Review Board of Chang Gung Memorial Hospital in Taiwan, which waived the requirement for informed consent due to anonymized data.

### 2.2. Data Collection and Statistical Analysis

Data were collected from the medical chart records, including precipitating factors, laboratory examinations, and clinical characteristics, such as onset time, initial Glasgow Coma Score, inotropic agents used, kidney or heart involvement, nosocomial infection, brainstem lesions, neurologic outcome, and mortality. Onset time was defined as the time at which neurological deficits first appeared. Post-treatment electroencephalograms (EEG) was categorized as either negative, or a focal/diffuse cortical dysfunction, or a focal, or multifocal or generalized epileptiform discharge. Cortical dysfunction was defined as slow waves in EEG [9]. Neurological outcomes at discharge were evaluated by a pediatric neurologist using the Pediatric Cerebral Performance Scale (PCPS) [10]. Neurological outcome scores were classified into two groups: (1) good outcomes (PCPS 1–3) and (2) poor outcomes (PCPS 4–6). Disease severity was also presented with ANE-SS (ANE severity score) [7] to compare the outcomes in children with ANE. The ANE-SS ranged from 0 to 9 points, with 3 points for shock at onset, 2 points for brainstem lesions, 2 points for age over 48 months, 1 point for plate count below 100,000/µL, and 1 point for CSF protein > 60 mg/dl. We classified patients into three groups according to ANE-SS: low risk (0–1 points), medium risk (2–4 points), and high risk (5–9 points).

### 2.3. Treatment

Pulse steroid therapy (methylprednisolone at 30 mg/kg/day for 3 days) and intravenous immunoglobulin (IVIG; 2 g/kg for 2–5 days) were administered to treat ANE. Early pulse steroid therapy was defined as administration within 24 h after the onset of ANE. 

### 2.4. Statstical Analysis

Categorical variables were analyzed using the chi-square test or Fisher’s exact test. The Mann-Whitney U test was used to compare continuous variables. Statistical significance was set at *p* < 0.05. All statistical analyses were conducted using IBM SPSS software, version 24 (IBM Corp., Armonk, NY, USA).

## 3. Results

Three patients did not receive pulse steroid therapy or immunoglobulin because of multiple organ failure and brain death on arrival at our hospital and were thus excluded. Therefore, a total of 26 patients were enrolled in the study. Table 1 shows the patients’ characteristics: the median age was 4 years (interquartile range, 2.3–8.4 years). There were no significant differences in age, sex, onset time, or nosocomial infection between survivors and non-survivors. The non-survivors had significantly higher rates of lower platelet count, inotropic agents used, brainstem involvement, and renal or heart impairment.

The overall mortality rate was 42.3%, and the non-survivors had significantly higher initial lactate and serum ferritin levels (*p* < 0.05). There were no statistically significant differences in laboratory data, including white blood cell count, C-reactive protein level, procalcitonin level, and cerebrospinal fluid analysis. Influenza was the most common pathogen, but there was no statistically significant difference in the rate of influenza between the two groups.

In general, there were no significant differences in the outcomes of pulse steroid therapy or pulse steroid plus immunoglobulin between survivors and non-survivors. When analyzing the time between symptom onset and usage of pulse steroid therapy, pulse steroid therapy used within 24 h after the onset of ANE resulted in significantly better outcomes (*p* = 0.039).

### Neurological Outcomes in Patients with and without Brainstem Involvement

In the group with brainstem involvement, 3 had good outcomes and 15 patients had poor outcomes (Table 2). Among the group with poor outcomes, 10 patients died, and five patients were diagnosed with brain death. In the group with good outcomes, two patients received IVIG alone and one received early pulse steroid therapy plus IVIG. There were two patients with medium risk and one patient with high risk in the group with good outcomes. Two of the three patients who received IVIG had good outcomes and were classified as medium risk. Six of the seven patients with brainstem lesions and early steroid use and most patients with mortality had multiorgan involvement and the use of inotropic agents (85–93%). Four patients had EEG recorded electrocortical silence, seven patients had cortical dysfunction with epileptiform discharge, and others lacked EEG records. However, whether patients received IVIG with or without early pulse steroid therapy had no significant impact on whether they had a good or poor outcome. Due to multiple organ failure and brain death on arrival at our hospital, three patients did not receive pulse steroid therapy or immunoglobulin. In the group with brain stem involvement, outcomes were not correlated with pulse steroid therapy, early pulse steroid therapy, or pulse steroid therapy combined with immunoglobulin. No major side effects of pulse steroid therapy or immunoglobulin usage were observed in this group.

In the group without brainstem involvement, seven patients had good outcomes and one patient had a poor outcome (Table 3). Among the group with good outcomes, four patients had no neurological sequelae, two patients had mild cognitive impairments, and one patient had mild cognitive impairment and motor deficits. One patient had ECG-recorded electrocortical silence; one patient had cortical dysfunction initially but recovered normal within months; five patients had no cortical dysfunction or epileptiform discharge. We classified three patients as low risk, four patients as medium risk, and one patient as high risk. Only one patient with a poor outcome was initially classified as low-risk. The majority of patients survived, including seven of the eight patients without brainstem lesions and early steroid use, without multiple organ involvement and use of inotropic agents (20–26%). All eight patients without brainstem involvement underwent early pulse steroid therapy, administered within 24 h after the onset of symptoms. No major side effects of pulse steroid therapy or immunoglobulin usage were observed in these patients.

As shown in Table 4, we compared the time to the administration of treatment and the dosage of steroid treatment between our study and that of and Mizuguchi [10]. In ANE without brainstem involvement, both studies reported higher percentages of good outcomes with early steroid usage compared to non-early steroid usage. Furthermore, in patients without brainstem involvement who received early steroid therapy, our study reported a higher percentage of good outcomes than Mizuguchi’s study when using a higher dose of steroids (pulse therapy: 87.5% vs. 58.3%, respectively). Unfortunately, neither study reported the benefits of early steroid therapy in patients with brainstem involvement.

## 4. Discussion

Our case review shows that early pulse methylprednisolone therapy may improve outcomes in children with ANE. The outcomes were excellent in ANE patients without brainstem involvement who were treated with pulse methylprednisolone therapy within 24 h after the onset of symptoms. Until now, the recommended dosage and duration of drug administration for ANE in pediatric groups have been controversial. Okumura et al. [11] first reported the efficacy of early steroid therapy against ANE. Our study also demonstrated the beneficial effects of early steroid treatment. Furthermore, we show that a high dose of methylprednisolone (pulse therapy) may have more benefits than a normal dose for patients with ANE without brainstem involvement. In the study of Okumura et al., 82.3% (14/17) of ANE patients without brainstem involvement received steroid therapy, with only three patients receiving pulse steroid therapy, but 66.6% (2/3) of the patients receiving pulse steroid therapy had a good outcome. In our study, all eight ANE patients without brainstem involvement received early pulse methylprednisolone therapy, and 87.5% (7/8) of the patients had good outcomes. A recent multicenter experience reported that in 12 patients receiving early pulse steroid therapy were associated with a good prognosis [12]. Therefore, pulse therapy (methylprednisolone at 30 mg/kg/day for 3 days) administered within 24 h after the onset of ANE may potentially be the optimal treatment for ANE.

In general, the clinical course of ANE presents in three stages: the prodromal stage, the acute encephalopathy stage, and the recovery stage [13]. In the prodromal stage, patients have a fever, upper airway tract symptoms, or gastroenteritis, which are caused by a viral infection. Acute encephalopathy, such as seizures or conscious disturbances, gradually appears within hours or days after the prodromal stage. In this stage, systemic inflammatory response syndrome has been observed in most patients with ANE, which may result from exaggerated immunity. At the recovery stage, most survivors have remaining neurological sequelae, while a few are able to recover completely. The most common hypothesis of the pathogenesis of ANE is hyper-cytokinemia resulting from an overreactive immune response induced by a viral infection, which then causes multiple organ impairment, including in the liver, kidney, heart, blood, and nervous system [6,14]. It has been indicated that inflammatory cytokines, including IL-6, tumor necrosis factor, and IFN-gamma, play an important role in the onset of ANE, and these cytokines are neurotoxic at high concentrations [6,15]. Okumura et al. considered that hyper-cytokinemia also results in brain injury through hyperpermeability of the blood–brain barrier and capillaries of the brain [11]. IL-10 is also elevated in patients with ANE, is a potent modulator of macrophage function, and is mainly produced by Th2 lymphocytes. Overexpression of IL-6 and tumor necrosis receptor factor-α may be produced by IL-10 through abnormal macrophage activation [16]. These cytokines and the hyperpermeability of the blood–brain barrier may cause necrosis of the vascular wall, resulting in vascular occlusion, after which necrotic tissue may lead to brain edema or hemorrhage [17,18]. 

Thus, we expect that immunomodulator therapy will effectively suppress the progression of a cytokine storm. Glucocorticoids can exhibit anti-inflammatory activity by binding cytoplasmic receptors, which regulate the transcription of anti-inflammatory genes [19,20], and glucocorticoid use has been considered as an anti-inflammatory therapy. Several studies have indicated that steroid, immunoglobulin, hypothermia therapy, and plasmapheresis might improve the outcomes of ANE [21,22,23,24]. The efficacy of corticosteroids has been reported to reduce mortality, hearing loss, and neurological sequelae in acute bacterial meningitis [25]. Although the efficacy of intravenous dexamethasone and pulse methylprednisolone therapy has not been systematically compared, a clear correlation was established between better prognoses and early steroid therapy (within 24 h) in ANE in patients without brainstem lesions [11,23]. Early immunomodulatory therapy is associated with fair outcomes in patients with ANE, as indicated by Appavu et al. [26]. Okumura et al. reported that, in 12 patients with ANE without brainstem lesions, 58% had no or mild neurologic sequelae after receiving early steroid therapy [11]. In our study, of eight patients who received pulse methylprednisolone therapy within 24 h of ANE onset without brainstem lesions, 87.5% presented better neurological outcomes. We chose pulse steroid therapy for ANE because we proposed that hyper-cytokinemia would be suppressed more efficiently by high-dose methylprednisolone. Increased blood interleukin 6 (IL-6) levels in the acute stage of encephalitis are associated with poor prognosis [27]. Therefore, recent studies have proposed that tocilizumab, a monoclonal antibody against the IL-6 receptor, may be a potential therapy for patients with ANE [28]. A previous study reported three patients who received tocilizumab, and two of these had a fair clinical and radiological recovery. Therapeutic hypothermia with immunotherapy was applied in a small case series and was associated with good outcomes [29]. Therapeutic hypothermia has an anti-cytokine effect that may be used in brain injury caused by cytokine storms. However, there was a small number of patients in these studies, and further clinical trials are needed to evaluate their safety and efficacy. In our study, pulse steroid therapy was safe and had no side effects such as hyperglycemia or opportunistic infection. Compared to previous studies, our study showed that early pulse methylprednisolone therapy in patients with ANE without brainstem lesions resulted in better prognoses than a normal dose of early steroid therapy.

Our study is limited by the small number of patients and the retrospective nature of the study at a single center; therefore, there are risks of missing data and information bias. Second, most patients with brainstem involvement were transferred from other hospitals. They did not receive any immunomodulatory therapy within 24 h after the onset of ANE until they were transferred to our hospital. Among the transferred patients, three of them had brain death when they arrived at our hospital. Brain-dead patients only received palliative treatment. Third, the incidence of ANE is so rare that randomized controlled trials will be difficult to initiate, although similar results have been reported in previous studies [7,21].

## 5. Conclusions

Our experience revealed that the use of early, high-dose methylprednisolone may be correlated with improved resolution of symptoms and improved clinical outcomes in patients without brainstem lesions. Further studies are needed to establish a consensus guideline for this fulminant disease.

## Figures and Tables

**Table 1 children-09-00136-t001:** Comparisons between survivors and non-survivors.

Demographics	All	Survivors	Non-Survivors	*p* Value
Number of patients	26	15	11	
Gender, male, *n* (%)	14 (53.8)	10 (66.7)	4 (36.3)	0.362
Age, years, median (IQR)	4.0 (2.3–8.4)	4.0 (2.5–11.3)	3.4 (2.3–7.4)	0.256
Inotropic agents used, *n*, (%)	14 (53.8)	4 (26.7)	10 (90.9)	0.001 *
Brainstem involvement, *n*, (%)	18 (69.2)	7 (46.7)	11 (100)	0.001 *
Kidney or heart impairment, *n*, (%)	12 (46.1)	3 (20)	9 (75)	0.002 *
Post–pulse therapy infection, *n*, (%)	4 (15.3)	3 (20)	1 (9)	0.316
Onset, days, median (IQR)	2.0 (1.0–3.0)	2.0 (1.0–3.0)	2.0 (1.0–3.0)	0.399
GCS, median (IQR)	8 (3–10)	9 (3–10.5)	5 (3–10)	0.221
**Laboratory examination, median (IQR)**				
WBC, µ/L	8900 (6150–13,900)	8400 (6300–11,100)	11700 (6300–11,100)	0.471
PLT,(x104/ µL)	19.3 (15.85–28.9)	21.2 (16–29)	18.6 (9.8–23.7)	0.049 *
ALT, U/L	113 (29–407)	55 (16–410)	135 (66.8–941)	0.159
AST, U/L	111 (31–490)	58 (30–350)	111 (50–1274)	0.162
Lactate, mg/dl	27.3 (13.0–50.5)	14.9 (9.3–30.9)	30.75 (13.3–65.8)	0.038 *
CRP, mg/dl	16.3 (6.1–42.7)	9.1 (4.9–19.8)	19.4 (13.7–50.4)	0.198
Procalcitonin, ng/mL	10.5 (2.8–61.2)	7.6 (2.1–47.6)	10.4 (5.3–69.7)	0.441
Ferritin, ng/ml	2726 (511.4–15,411)	816.4 (356.9–2973.5)	9169 (1379–24,288)	0.022 *
CK(U/L)	233.5 (158.5–289)	226(118–433)	405(162–1153)	0.454
CSF lactate, mg/dl	21.2 (18.3–37.6)	20.0 (17.5–22.1)	37.6 (18.9–118.6)	0.201
CSF leukocyte, µ/L	3.5 (1.0–15.5)	1 (0–7)	6 (3–310)	0.062
CSF protein, mg/dl	66.3 (38.3–460.1)	64.9 (30.7–111.1)	460.1 (32.6–937.5)	0.086
Pathogen, *n* (%)				
Influenza A	10 (38.4)	5 (33.3)	5 (45.4)	0.885
Influenza B	3 (11.5)	3 (20)	0	0.088
Mycoplasma	5 (19.2)	2 (13.3)	3 (27.2)	0.564
Unknown	11 (42.3)	5 (33.3)	6 (54.5)	
ANE-SS risk, *n* (%)	26	15	11	<0.001 *
ANE-SS (0–1) low risk, *n* (%)	3 (11.5)	3 (20)	0	
ANE-SS(2–4) medium risk, *n* (%)	7 (26.9)	7 (46.7)	0	
ANE-SS(5–9) high risk, *n* (%)	16 (61.5)	5 (33.3)	11 (100)	
Treatment, *n* (%)				
Pulse steroid therapy	21 (80.7)	12 (80)	9 (81.8)	0.344
Pulse steroid + IVIG	15 (57.6)	8 (53.3)	7 (63.6)	0.858
Early pulse steroid therapy	15 (57.6)	8 (53.3)	7(63.6)	0.039 *

* Statistically significant at *p* < 0.05. Abbreviations: IQR = interquartile range; GCS = Glasgow Coma Scale; WBC = white blood cell; ALT = alanine aminotransferase; AST = aminotransferase; CRP = C-reactive protein; CSF = cerebrospinal fluid; ANR-SS = acute necrotizing encephalopathy severity score; IVIG = intravenous immunoglobulin.

**Table 2 children-09-00136-t002:** Neurologic outcome and ANE-SS in patients with brainstem involvement.

	All	Low Risk (ANE-SS 0–1)	Medium Risk (ANE-SS 2–4)	High Risk (ANE-SS 5–6)	*p* Value	Good (PCPS 1–3)	Poor (PCPS 4–6)	*p* Value
All patients, *n* (%)	18	0	3	15		3	15	
Pulse steroid therapy, *n* (%)	13	0	1 (33.3)	12 (80)	0.546	1 (33.3)	12 (80)	0.271
IVIG alone used, *n* (%)	3	0	2 (66.7)	1 (6.6)	0.019	2 (66.7)	1 (6.6)	0.06
Pulse steroid + IVIG, *n* (%)	10	0	1 (33.3)	9 (60)	0.865	1 (33.3)	9 (60)	0.593
Early pulse steroid therapy, *n* (%)	7	0	1 (33.3)	6 (40)	1	1 (33.3)	6 (40)	0.726
No IVIG or pulse steroid therapy, *n* (%)	3	0	0	3 (20)	0.748	0	3 (20)	0.445

ANR-SS = acute necrotizing encephalopathy severity score; PCPS = Pediatric Cerebral Performance Scale; IVIG = intravenous immunoglobulin.

**Table 3 children-09-00136-t003:** Neurological outcome and ANE-SS in patients without brainstem involvement.

	All	Low Risk (ANE-SS 0–1)	Medium Risk (ANE-SS 2–4)	High Risk (ANE-SS 5–6)	*p* Value	Good (PCPS 1–3)	Poor (PCPS 4–6)	*p* Value
All patients, *n* (%)	8	3	4	1		7	1	
Pulse steroid alone therapy , *n* (%)	3	1 (33.3)	1 (25)	1 (100)	0.375	2 (28.5)	1 (100)	0.77
Pulse steroid + IVIG, *n* (%)	5	2 (66.7)	3 (75)	0	0.375	5 (71.4)	0	0.77
Early steroid therapy, *n* (%)	8	3 (100)	4 (100)	1 (100)	1	7 (100)	1 (100)	<0.001

ANR-SS = acute necrotizing encephalopathy severity score; PCPS = Pediatric Cerebral Performance Scale; IVIG = intravenous immunoglobulin.

**Table 4 children-09-00136-t004:** Comparison of outcomes in pulse steroid or early steroid use in ANE, with or without brainstem involvement [7].

Outcome	Our Study			Mizuguchi’s Study [7]		
ANE without brainstem involvement	Early pulse steroid	Non-early pulse steroid	*p* value	Early steroid	Non-early steroid	*p* value
	(*n* = 8)	(*n* = 0)		(*n* = 12)	(*n* = 5)	
Good, *n* (%)	7 (87.5)	0	<0.01	7 (58.3)	0	0.044
Poor, *n* (%)	1 (12.5)	0		5 (41.7)	5 (100)	
ANE with brainstem involvement	(*n* = 7)	(*n* = 14)		(*n* = 9)	(*n* = 8)	
Good, *n* (%)	1 (14.2)	2 (14.2)	0.51	0	2 (25)	0.39
Poor, *n* (%)	6 (85.8)	12 (85.8)		9 (100)	6 (75)	

ANE = acute necrotizing encephalopathy.

## Data Availability

The data used to support the findings of this study are available from the corresponding author upon request.

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
