# Peer review of "Early High-Dose Methylprednisolone Therapy Is Associated with Better Outcomes in Children with Acute Necrotizing Encephalopathy"

_children, 2022, doi:10.3390/children9020136_

Round 1

Reviewer 1 Report

Summary:

This manuscript investigates a therapeutic efficacy of steroid and immunomodulatory administrations for acute necrotizing encephalopathy (ANE) in children by reviewing the medical chart retrospectively to identify better treatment strategies. I address some major concerns that should be addressed.

Major concerns:

Study design and abstract:

  • The authors clearly declared the study design of this investigation as a retrospective review. If so, the word “cohort” would be misuse or is likely to make readers confusing, as it typically refers to prospective analysis. The authors should check and correct whether this word should be replaced in all this manuscript.
  • Moreover, the description in background is only demographical and far from the aim of this study. It should be described in a straightforward manner.

Methods:

  • The reference for the PCPS needs to be cited properly.
  • Are there any reasons for not using analysis of covariance in the statistical analysis of this study, which seems to be better?

Discussion:

  • The descriptions are rather redundant and are not written in a convergent manner using topic sentences effectively. Moreover, which findings are novel and how those are consistent with previous studies are not clearly mentioned. From a clinical standpoint, the authors should discuss pros and cons of early steroid therapy and clinical time course even if it would be difficult to show clearly. In other words, the authors should discuss the most important findings in a straightforward way.

Conclusion

The correlation between the steroid therapy and the treatment outcome may be overstatement since any correlation analyses was not performed in this study.

.

In summary, the authors should reconsider these major points to make the study rationale and show how it can be novel and informative based on the chart data retrospectively in a simple and convergent manner, together with refinement of writing in terms of clarity, coherence, and capacity.

Minor points

Tables:

There are quite many misalignments, unnatural font size, misuses of bold or italic, redundancy. The authors should refine the tables extensively since even this highly lax presentations can be a rational reason of reject and make the responsibility of corresponding author questionable.

For example, and specifically, in Table 1, I think that the title Survival and Mortality need to be replaced with survivors and non survivors; “Initial” is quite redundant.

Ethical considerations:

There is an incongruent ethical considerations, that is, nationality between the institutional review board (Seoul) and the affiliation of the authors (Taiwan). The authors should show the reason for this discrepa

Author Response

Response to Reviewer 1 Comments

Point 1: The authors clearly declared the study design of this investigation as a retrospective review. If so, the word “cohort” would be misuse or is likely to make readers confusing, as it typically refers to prospective analysis. The authors should check and correct whether this word should be replaced in all this manuscript.

Response 1: Thanks for your comments sincerely.

The word “cohort” was misuse and were all deleted in this manuscript in line 16、94、164、170、171、174、176、190、194 and table 4.

Point 2: Moreover, the description in background is only demographical and far from the aim of this study. It should be described in a straightforward manner.

Response 2: Thanks for your comments sincerely.

We had added the statement on the page 1 line 13 to 14 as: Steroid and immunoglobulin treatments have been the most commonly used treatment options for ANE, but their therapeutic efficacy is still controversial.

Point 3: The reference for the PCPS needs to be cited properly

Response 3: Thanks for your comments sincerely.

We had added the reference 10 for the citation of PCPS.

Reference 10: Pollack, M.M.; Holubkov, R.; Funai,T.; Clark,A.; Moler,F.; Shanley, T.; Meert,K.; Newth,C.J.L.; Carcillo,J.; Berger, J.T. et al. Relationship Between the Functional Status Scale and the Pediatric Overall Performance Category and Pediatric Cerebral Performance Category Scales FREE. JAMA Pediatr. 2014, 168, 671–676.

Point 4: Are there any reasons for not using analysis of covariance in the statistical analysis of this study, which seems to be better?

Response 4: Thanks for your comments sincerely.

Due to relative small case numbers, there is problem about overfitting if putting more than 2 parameters in the logistic regression model. So we put two import parameters (early pulse steroid therapy and IVIG) to this logistic regression model as the following:  

The multivariate logistic regression model reported that the early pulse steroid therapy

was associated with good neurologic outcome in ANE without brainstem involvement.

Table. Multivariate logistic regression model to predict outcome (PCPS≦3) at discharge

Group

Parameters

Adjusted Odds ratio

p-value

ANE with brainstem involvement

Early pulse steroid therapy

2 (0.15-26.19)

0.597

IVIG

1.25 (0.971-1.61)

0.622

ANE without brainstem involvement

Early pulse steroid therapy

0.143 (0.09-0.85)

0.039*

IVIG

0.667 (0.3-1.484)

0.783

Point 5: The descriptions are rather redundant and are not written in a convergent manner using topic sentences effectively. Moreover, which findings are novel and how those are consistent with previous studies are not clearly mentioned. From a clinical standpoint, the authors should discuss pros and cons of early steroid therapy and clinical time course even if it would be difficult to show clearly. In other words, the authors should discuss the most important findings in a straightforward way.

Response 5: Thanks for your comments sincerely.

The important findings in a straight way are mentioned as the following:

Page 6 line 182 to 185: Our case review shows that early methylprednisolone pulse therapy may improve outcomes in children with ANE. The outcomes were excellent in ANE patients without brainstem involvement who were treated with methylprednisolone pulse therapy within 24 hours after the onset of symptoms.

Page 6 line 194 to 197: In our study, all 8 ANE patients without brainstem involvement received early methylprednisolone pulse therapy, and 87.5% (7/8) of the patients had good outcomes. A recent multicenter experience reported that 12 patients receiving early pulse steroid therapy was associated with a good prognosis [12]. Therefore, pulse therapy (methylprednisolone at 30 mg/kg/day for 3 days) administered within 24 h after the onset of ANE may be the optimal treatment for ANE potentially.

For the clinical standpoint about pros and cons of early steroid therapy and clinical course, we mentioned as the following:

Page 7 line 228 to 231: Although the efficacy of intravenous dexamethasone and methylprednisolone pulse therapy has not been systematically compared, a clear correlation was established between better prognoses and early steroid therapy (within 24 hours) in ANE in patients without brainstem lesions [11,24]. Early immunomodulatory therapy is associated with fair outcomes in patients with ANE, as indicated by Appavu et al. [27]. Okumura et al. reported that in 12 patients with ANE without brainstem lesions, 58% had no or mild neurologic sequelae after receiving early steroid therapy [11]. In our study, of 8 patients who received methylprednisolone pulse therapy within 24 hours of ANE onset without brainstem lesions, 87.5% presented better neurologic outcomes. We chose steroid pulse therapy for ANE because we proposed that hypercytokinemia would be suppressed more efficiently by high-dose methylprednisolone. Increased blood interleukin 6 (IL-6) levels in the acute stage of encephalitis are associated with poor prognosis [28].

Page 7 line 248 to 251: In our study, steroid pulse therapy was safe and had no side effects such as hyperglycemia or opportunistic infection. Compared to previous studies, our study showed that early methylprednisolone pulse therapy in patients with ANE without brainstem lesions resulted in better prognoses than a normal dose of early steroid therapy

Point 6: The correlation between the steroid therapy and the treatment outcome may be overstatement since any correlation analyses was not performed in this study.

Response 6: Thanks for your comments sincerely.

It is true that the correlation between early pulse steroid therapy and outcome may be overstatement based on the small number cases. Therefore we deleted the statement on page 7 line 264 to 266 as: In conclusion, methylprednisolone pulse therapy administered within 24 hours after the onset of symptoms is potentially optimal for the treatment of ANE without brainstem involvement in children. 

Added the statement on page 7 line 266 to 267 as: Further studies are needed to establish a consensus guideline for this fulminant disease.

Page 1 line 29 to 31 as: Pulse steroid therapy (methylprednisolone at 30 mg/kg/day for 3 days) administered within 24 hours after the onset of ANE may be correlated with good prognosis. Further studies are needed to establish a consensus guideline for this fulminant disease.

Point 7: There are quite many misalignments, unnatural font size, misuses of bold or italic, redundancy. The authors should refine the tables extensively since even this highly lax presentations can be a rational reason of reject and make the responsibility of corresponding author questionable.

For example, and specifically, in Table 1, I think that the title Survival and Mortality need to be replaced with survivors and non survivors; “Initial” is quite redundant.

Response 7: Thanks for your comments sincerely.

We have revised all the four tables according to your professional advice.

Point 8: There is an incongruent ethical considerations, that is, nationality between the institutional review board (Seoul) and the affiliation of the authors (Taiwan). The authors should show the reason for this discrepa.

Response 8: Thanks for your comments sincerely.

The previous information about IRB was wrong. We have revised the correction information on page 8 line 276 to 278 as: Institutional Review Board Statement: The study was conducted according to the guidelines of the Declaration of Helsinki and was approved by the Institutional Review Board of Chang Gung Memorial Hospital in Taiwan. (protocol code: 201900095B0; date of approval: 29 November 2020).

Reviewer 2 Report

Title add IVIG "Early high-dose methylprednisolone and IVIG therapy is associated with better outcomes in children with acute necrotizing encephalopathy"

Methods:

  • "pulse steroid therapy" is the most used term in literature not  "steroid pulse therapy"
  • What does EEG cortical dysfunction mean? should be written in the method with a reference.

Results:

  • p small letter not P Capital letter as statistical significane
  • "Due to multiple organ failure and brain death when arriving at our hospital, three patients did not receive steroid pulse therapy or immunoglobulin."  They should be excluded from the study as they arrived with multiorgan failure prior to any intervention. Then only 26 children to be included.
  • "As shown in Table 4, we compared the time to administration of treatment and the 159 dosage of steroid treatment between our cohort and Mizuguchi’s cohort [9]." 

Discussion :

  • Bashiri et al 2020, study can be used 
    Acute Necrotizing Encephalopathy of Childhood: A Multicenter Experience in Saudi Arabia.

https://www.frontiersin.org/articles/10.3389/fped.2020.00526/full

Author Response

Response to Reviewer 2 Comments

Point 1: "pulse steroid therapy" is the most used term in literature not "steroid pulse therapy"

Response 1: Thanks for your comments sincerely.

We have revised all the sentences of “steroid pulse therapy” to "pulse steroid therapy" in the manuscript.

Point 2: What does EEG cortical dysfunction mean? should be written in the method with a reference.

Response 2: Thanks for your comments sincerely.

We have added the statement as the following on page 2 line 79 to 82 as: Post-treatment electroencephalogram (EEG) were categorized as having either negative, or a focal/diffuse cortical dysfunction, or a focal, or multifocal or generalized epileptiform discharge. The cortical dysfunction was defined as slow waves in EEG [9].

Point 3: p small letter not P Capital letter as statistical significane

Response 3: Thanks for your comments sincerely.

We have revised all the P Capital letter as p small letter in the manuscript.

Point 4: "Due to multiple organ failure and brain death when arriving at our hospital, three patients did not receive steroid pulse therapy or immunoglobulin."  They should be excluded from the study as they arrived with multiorgan failure prior to any intervention. Then only 26 children to be included.

Response 4: Thanks for your comments sincerely.

We have revised that a total of 26 patients were enrolled in the study.

Point 5: Bashiri et al 2020, study can be used.
Acute Necrotizing Encephalopathy of Childhood: A Multicenter Experience in Saudi Arabia.

Response 5: Thanks for your comments sincerely.

The important findings in a straight way are mentioned as the following:

We have added Bashiri’s study on page 6 line 196 to 197 as: A recent multicenter experience reported that 12 patients receiving early pulse steroid therapy was associated with a good prognosis [12].

Round 2

Reviewer 1 Report

I think that the revised manuscript has reached to an acceptable level of quality. Thank you for your corrections.